# A Case Study in Attention-Deficit/Hyperactivity Disorder: An Innovative Neurofeedback-Based Approach

**DOI:** 10.3390/ijerph19010191

**Published:** 2021-12-24

**Authors:** Paloma Cabaleiro, Marisol Cueli, Laura M. Cañamero, Paloma González-Castro

**Affiliations:** Department of Psychology, University of Oviedo, 33003 Oviedo, Spain; uo267227@uniovi.es (P.C.); lauramcanamero@uniovi.es (L.M.C.); mgcastro@uniovi.es (P.G.-C.)

**Keywords:** ADHD, attention, neurofeedback, theta/beta protocol, SMR protocol

## Abstract

In research about attention-deficit/hyperactivity disorder (ADHD) there is growing interest in evaluating cortical activation and using neurofeedback in interventions. This paper presents a case study using monopolar electroencephalogram recording (brain mapping known as MiniQ) for subsequent use in an intervention with neurofeedback for a 10-year-old girl presenting predominantly inattentive ADHD. A total of 75 training sessions were performed, and brain wave activity was assessed before and after the intervention. The results indicated post-treatment benefits in the beta wave (related to a higher level of concentration) and in the theta/beta ratio, but not in the theta wave (related to higher levels of drowsiness and distraction). These instruments may be beneficial in the evaluation and treatment of ADHD.

## 1. Introduction

Attention-deficit/hyperactivity disorder (ADHD) is one of the most common childhood disorders, affecting between 5.9% and 7.2% of the infant and adolescent population. The fifth edition of the Diagnostic and Statistical Manual of Mental Disorders [1] describes ADHD as a neurodevelopmental disorder characterized by a persistent pattern of inattention, hyperactivity, and impulsivity manifesting in children before the age of 12 years old more frequently and with greater severity than expected in children of equivalent ages. Depending on the predominant symptoms, three types of presentation may be identified: predominantly hyperactive-impulsive, predominantly inattentive, and combined. There are two theories that attempt to explain the neurophysiological nature and characteristics of ADHD. Mirsky posited a deficit in attention as the main focus in ADHD, such that the failure is found in processes of activation [2]. The other theory was proposed by Barkley, who attributed the problems of ADHD to a deficit in behavioral regulation, where processes associated with the frontal cortex fail [3].

The determination of ADHD symptoms, along with the underlying neuropsychology, as outlined by the theories above, have led in recent years to the incorporation of evaluation and intervention techniques that do not solely focus on the behavioral aspects of the disorder. More specifically, techniques such as electroencephalography in ADHD evaluation and neurofeedback in interventions may provide greater benefits in detection and treatment.

The present study analyzes a specific case of ADHD with predominantly inattentive presentation, covering monopolar electroencephalogram recording (brain mapping called MiniQ) and intervention via neurofeedback.

The study was approved by the relevant Ethics Committee of the Principality of Asturias (reference: PMP/ICH/135/95; code: TDAH-Oviedo), and all procedures complied with relevant laws and institutional guidelines.

### 1.1. Evaluation of ADHD

The current diagnostic criteria for ADHD can be found in the DSM-5 [1] and in the International Statistical Classification of Diseases and Related Health Problems, eleventh revision, from the World Health Organization [4]. Various evaluation instruments are used to identify ADHD, from general assessments via broad scales such as the Wechsler scale, to more specific tests assessing execution (e.g., test of variables of attention, D2 attention test), symptoms (e.g., Conners scale, EDAH scale), and the evaluation of cortical activity (e.g., using quantitative electroencephalograms, qEEG).

One alternative to qEEG is monopolar EEG recording (fundamentally used in clinical practice), called MiniQ (software Biograph Infinity, ThoughtTech, Montreal, QC, Canada). The MiniQ is an instrument for evaluating brain waves from 12 cortical locations (international 10/20 system) [5]. This type of evaluation (monopolar EEG, MiniQ) lies somewhere between the traditional baseline (single-channel qEEG) and full brain mapping. The frequency ranges evaluated match the classics [6,7]: delta 1–4 Hz, theta 4–8 Hz, alpha 8–12 Hz, sensorimotor rhythm SMR 12–15 Hz, beta 13–21 Hz, beta3 or high beta 20–32 Hz, and gamma 38–42 Hz. Theta waves have been related to low activation, sleep states, and low levels of awareness, beta and alpha waves have been associated with higher levels of attention and concentration [8]. In addition, the MiniQ, in line with qEEG, provides the relationships or ratios of theta/alpha, theta/beta, SMR/theta and peak alpha. Previous research has established that the ratio between theta and beta waves is a better indicator of brain activity than each wave taken separately (see Rodríguez et al. [9]). Monastra et al. attempted to establish what values of the theta/beta ratio would be compatible with those seen in subjects with ADHD [7]. They indicated critical values (cutoff points) for ADHD in theta/beta absolute power ratio, using 1.5 standard deviations compared to the control groups and based on age, those cutoff points are: 4.36 (6–11 years old), 2.89 (12–15 years old), 2.24 (16–20 years old), and 1.92 (21–30 years old). Higher values than the cutoff points would indicate a profile that is compatible with a subject with ADHD.

The distribution of electrical brain activity must be analyzed considering each site and the expected frequency. A regulated subject is characterized by more rapid activity in the frontal regions (predominantly beta) which decreases toward the posterior (occipital) regions, where slower waves (theta and delta) are expected [10,11]. Slower brainwaves are expected to predominate in the right hemisphere compared to the left, in which faster waves predominate. More specifically, beta waves will predominate in the left hemisphere, alpha waves in the right hemisphere, and there will be similar levels of theta waves in both. In addition, during a task (e.g., reading or arithmetic) rapid (beta) waves are expected to increase.

In contrast, the electrical activity in a subject with predominantly inattentive ADHD is characterized by a predominance of theta waves (compared to beta) in the frontal regions, particularly on the left (F3). During tasks (e.g., reading or arithmetic), a subject with predominantly inattentive ADHD will exhibit increased slower (theta) waves, and there will be a slowdown in the frontal regions that hinders attentional quality, as suggested by researchers such as Clarke et al. [10] and more recently, Kerson et al. [12]. Studying the profile of cortical activation allows suitable intervention protocols to be established and tailored to each subject.

### 1.2. ADHD Intervention

Many studies have examined the efficacy of the various treatments and interventions aimed at improving symptoms associated with ADHD (inattention, hyperactivity, and impulsivity), such as medication, behavioral treatments, and neurofeedback (see Caye et al. [13]). Neurofeedback is a type of biofeedback which aims for the subject to be aware of their brain activity and to be able to regulate it via classical conditioning processes [14,15]. In neurofeedback training, a subject’s electrical brain activity is recorded via an electroencephalograph, and the signal is filtered and exported to a computer. Software then transforms and quantifies the brainwaves, presenting them in the form of a game with movement or sounds which give the subject feedback about their brain activity [16].

The use of neurofeedback in interventions for ADHD began in 1973, although the first study with positive results was published in 1976 [17]. Since then, various studies have reported benefits from using neurofeedback in infants, with improvements in behavior, attention, and impulsivity control (e.g., [18,19,20,21,22]). A meta-analysis by Arns et al. [14] concluded that treatment of ADHD with neurofeedback could be considered “effective and specific”, with a large effect size for attention deficit and impulsivity and a moderate effect size for hyperactivity. In a systematic review and meta-analysis, Van Doren et al. [21] found that neurofeedback demonstrated moderate benefits for attention and hyperactivity-impulsivity, which were maintained in subsequent follow-ups (between 2 and 12 months after the intervention). However, in a recent meta-analysis aimed at comparing the effects of methylphenidate and neurofeedback on the main symptoms of ADHD, Yan et al. [20] found methylphenidate to be better than neurofeedback, although the authors highlighted that the results were inconsistent between evaluators.

Neurofeedback training is normally done two or three times a week, and around 40 sessions are needed to see changes in symptomatology [13]. Although it is an expensive treatment that needs consistency and continuity, in the USA, around 10% of children and adolescents with ADHD have received neurofeedback [23]. The benefits of neurofeedback training may depend on the type of protocol used. The three most-commonly used protocols in subjects with ADHD are [14]: (1) theta/beta ratio; (2) sensorimotor rhythm, SMR; and (3) slow cortical potential. The most widely used of these three protocols is the theta/beta ratio, based on inhibition of theta and increasing beta, which usually improves SMR at the same time [13]. However, it is important to note that there is no recommended standard about the number, time or frequency of sessions, and there is no standard placement of NF screening when this type of protocol is administered [24,25]. In this context, the present study aims to provide a structure in which the neurofeedback intervention is adjusted based on the data provided by the previous assessment in a specific case.

The intervention protocol must be tailored to each individual case based on prior assessment, especially when using results from tests such as the MiniQ. In this context, the objective of the current study is to present the process of analyzing brainwaves in a case with ADHD (predominantly inattentive presentation) via the MiniQ test, the protocol for intervention using neurofeedback, and its efficacy. Although the alteration of brainwaves in specific areas in subjects with ADHD is well documented, and the efficacy of neurofeedback has been observed in various studies, the present study aims to provide a specific procedure for assessment and intervention. Researchers and professionals need specific protocols and procedures that allow them to determine what is effective for each individual case.

## 2. Methodology

### 2.1. Description of the Case

This was a case study using monopolar electroencephalogram recording (brain mapping known as MiniQ) for subsequent use in an intervention with neurofeedback for a 10-year-old girl presenting predominantly inattentive ADHD.

#### 2.1.1. Patient Identification

The subject was a 10-year-old girl in the fourth year of primary education. Her academic performance was poor, with the worst results in language, social sciences, and science. She found it difficult to go to school and was shy and reserved. She was the younger of two sisters, the older being an outstanding pupil. Her mother characterized her as a quiet girl who needed a lot of time to do any kind of task. In addition, during the study and academic tasks, she would often gaze into space, as if she were in her own world. Both her father and her mother evidenced concern for her school results, but also for her social relationships, as her self-absorption appeared in all contexts, making it hard for her to have conversations, pay attention to others, or follow the rules in games.

#### 2.1.2. Reason for Consultation

The consultation was for poor academic performance, slowness doing tasks, and wandering attention from when she had started school, although that had increased in the previous year. Initially, the subject did not demonstrate any great willingness to attend the consultations, but over time, she demonstrated a participative attitude with good involvement in doing the tasks she was set.

#### 2.1.3. History of the Problem

The subject’s school history was one of failure in the main school subjects. She had not had to repeat a school year, but her form tutors repeatedly raised this possibility with her parents. At the time of the study, there had been no clinical or educational psychology assessments. Previous diagnosis of ADHD was by her neuropediatrician one month before the assessment in the Psychology clinic consultation. From that point, guidelines were given for pharmacological treatment, which had not begun.

### 2.2. Proposed Evaluation and Intervention

#### 2.2.1. Evaluation: Brainwave Analysis with the MiniQ Instrument

An assessment was performed using a MiniQ (Monopolar, from Biograph Infinity). Assessment using the MiniQ is a two-step process (evaluation and interpretation) which is simple, relatively fast, and inexpensive.

The first step is to make the recording from the 12 cortical sites, which can be done with eyes closed or open, and either with or without tasks (reading or arithmetic). This gives information about the values of the different brainwaves at each site. To begin, electrodes are placed on the earlobes and two active electrodes in each of the sites indicated by the program. Before beginning the assessment for each site, the impedance level—the quality of the connection—for each of the electrodes must be checked, both on the ears and on the scalp, to avoid artefacts. When the impedance level is below 4, the recording process can begin. The subject is instructed to remain still and to look at the computer screen where there is an image of a landscape. They must keep their eyes open and keep silent. The program guides the application, which is based on the placement of electrodes in groups of two following the sequence: Cz–Fz, Cz–Pz, F3–F4, C3–C4, P3–P4, O1–O2, and T3–T4. For sites F3–F4, subjects are asked to read a story quietly and to do some simple arithmetic (e.g., 2 + 3, +5, +4, −1, +6, −3, etc.). Once recordings have been made at all of the sites, the program filters the data to remove artefacts. Finally, the recorded data is interpreted, and the values are analyzed, allowing the state of the subjects’ brainwaves to be determined. Applying the test takes approximately 60 min.

The second step is to analyze the collected data considering the site and the frequency ranges at each. The sites are labelled based on the four quadrants of the cortex: anterior, posterior, left hemisphere (odd numbers), and right hemisphere (even numbers). The instrument gives the results in two formats, an Excel spreadsheet and a PowerPoint. In addition to the measurements or wave values (delta, theta, alpha, sensorimotor rhythm SMR, beta, beta3, and gamma) at the sites noted above, the spreadsheet also includes the values for the ratios of theta/alpha, theta/beta, SMR/theta, and peak alpha. The PowerPoint presentation gives the same information, although over a background image of a brain, which allows scores to be seen at the relevant site (see Figure 1). With that information, it is possible to assess cerebral asymmetry, both anterior-posterior and right-left, according to each location.

The values of the theta/beta ratios are interpreted based on Monastra et al. (1999) [7], bearing in mind that in this case, the scores were relative power not absolute. Scores are indicative of ADHD when the values are over 2.5 for those up to 7 years old, over 2.8 for 7- to 11-year-olds, over 2.4 in adolescents, and over 1.8 in adults. Traditional ratios for ADHD indicators use absolute power values measured in peak volts (microvolts squared divided by the hertz value). Biograph for theta/beta ratio calculation uses relative power values (microvolts divided by the hertz value).

#### 2.2.2. Intervention: Neurofeedback Protocols

The intervention was carried out using the Biograph Infiniti biofeedback software (Procomp2 from Thought Technology, Montreal, QC, Canada; https://thoughttechnology.com/, accessed on 23 December 2021). Two protocols were used in the intervention process, an SMR protocol and a theta/beta protocol. The protocol and specific sites selected were based on the prior evaluation.

The SMR protocol used site Cz and was designed to work on three frequencies, theta, SMR, and beta3 [26]. The objective of this kind of protocol is to perform SMR (12–15 Hz) training to increase the production of this wave and inhibit the production of theta (4–7 Hz) and beta3 (20–32 Hz) activity. During the training sessions, the subject watches a videogame or a film on the screen. Following the neurofeedback dynamic, the game or the film progresses positively if the level of electrical activity increases and stops when the level of electrical activity falls. Reinforcement occurs when the value of theta and beta3 are below the set value and SMR is above a pre-determined threshold. The reinforcement consists of a sound and points awarded to the subject. The working thresholds are provided by the program automatically, although they can be modified manually by the therapist. The level of reinforcement is set by the therapist. Initially, it is set at 80%, and depending on how the subject masters the task, the reinforcement is reduced. The subject is not given explicit instructions about what they have to do; they are told “try to keep the animation on the screen moving”.

The theta/beta protocol works at site Fz. The aim of this protocol is to reduce the amplitude of theta waves and increase beta to work on concentration. The subject has to do tasks which consist of concentrating on a game that appears on the computer screen. The game presents a pink square (which represents the value of theta) and a blue square (representing the value of beta). The subject is told that the game involves trying to make the pink square as small as possible and the blue square as large as possible. The computer automatically generates the ranges over which the waves are worked, although they can be changed manually by the therapist. The desired working theta/beta ratio can also be set manually. The protocol begins with high ratios, close to three, such that the task is simple and the subject achieves reinforcement on many occasions. The ratio is progressively reduced according to the subject’s progress.

The intervention lasted for a year and consisted of 75 neurofeedback sessions. There were two phases to the training. The first phase, “the regulation phase”, covered the first 15 sessions, during which the SMR protocol was followed at Cz. The aim of this first phase was to strengthen SMR and inhibit theta and beta3 in the central region. These sessions were around 45 min each. To avoid tiredness, different presentations of neurofeedback were used (videogame or film) during the sessions, with five-minute breaks between each presentation.

The second phase ran from session 16 to session 75. In these sessions, the SMR protocol at Cz was applied for 20 min, followed by a five-minute break before the theta/beta protocol at Fz was applied for another 20 min. For the first six months of the intervention, sessions were 45 min, twice weekly. During the remaining six months, the sessions were weekly and remained 45 min long.

## 3. Results

### 3.1. Brainwave Evaluation

Based on the information obtained over the evaluation of the case, and considering the prior diagnosis from her pediatric neurologist, the subject presented ADHD with predominantly inattentive presentation. As Figure 1 shows, her brainwave profile indicated scores for the theta/beta ratio of close to 2.8 in the central (Cz) and frontal regions (Fz). Considering the scores in Cz and Fz, the neurofeedback needed to include these sites. Furthermore, neurofeedback on frontal-midline theta (Fz) has been shown to be frequently more effective than neurofeedback protocols that do not include Fz [22].

Given the brainwave profile, the aim of the intervention was to reduce theta and increase beta in the frontal zones. That indicated using the SMR and theta/beta protocols [15].

### 3.2. Progression following Neurofeedback Intervention

Once the neurofeedback intervention was completed, brainwave activity was assessed again using the MiniQ. Figure 2 illustrates the change in theta, beta, and SMR, along with the theta/beta ratio at sites Cz and Fz. The results show a positive progression following the neurofeedback training.

Theta activity fell following the intervention, both at Fz (by 0.77) and at Cz (by 1.56). To put it another way, there was a reduction in the slow wave at both sites (mainly in the central region compared to the frontal region). This is in line with expected values of theta at the cortical level, as they should be higher in posterior areas and lower in frontal areas.

There was also an increase in beta at the two sites, with a 3.60-point increase at Fz and a 4.2-point increase at Cz. In this case, the intervention produced considerable increases in the rapid wave values at both sites, although the value was slightly higher in the central area than in the frontal. Values for beta waves are expected to be higher in frontal areas than central areas, and although that was not the case here, the values were very close. The SMR wave also increased notably, by 2.57 points at Fz and 2.89 points at Cz. In short, the intervention led to a slight reduction in the slow wave, with lower values at post-treatment (less distraction), and increases in fast waves, beta, and SMR, with higher values after the intervention (better ability to concentrate). The theta/beta ratio also decreased at post-treatment (basically due to the increase in beta), both at Fz (by 0.69) and Cz (by 0.96), from values close to those for ADHD to scores more indicative of a subject without ADHD.

In addition, as initially proposed, the assessment with the MiniQ also considered the subject’s activation levels during reading and arithmetic tasks. Measurement of these values was at sites F3 and F4. The subject did three types of task for two minutes each: Paying attention to the screen on which a landscape appeared, reading a story, and doing simple arithmetic (addition and subtraction). As Figure 3 shows, post-treatment scores were different than pre-treatment scores.

In the first task (pay attention to the screen), the values for theta, beta, and beta3 at F3 and F4 all rose. In the second and third tasks (reading and arithmetic), there were variations in all of the waves, both slow and fast. These results indicate that there was no improvement during tasks following the intervention, because although the fast waves (beta and beta3) increased, the slow wave (theta) did not diminish. Following the intervention, the expectation was to have increased levels of beta and beta3 (especially at F3), while reducing levels of theta. However, as Figure 3 shows, the theta/beta ratio fell, with lower values post-treatment.

## 4. Discussion

The aim of this study was to present the process for detecting a case of ADHD (predominantly inattentive presentation) using the MiniQ test, along with the neurofeedback intervention protocol and its efficacy. In terms of detection, the MiniQ showed the subjects’ brain activity, which together with behavioral symptoms, provided details of their characteristic profile and allowed tailored treatment. Various studies in the literature have concluded that children with ADHD exhibit higher levels of theta waves and lower levels of beta waves, particularly in frontal areas [10,11]. In addition, the relationship between the theta and beta waves (the theta/beta ratio) had already been associated with ADHD symptomatology through the research by Monastra et al. [7] and Jarrett et al. [27].

In the current case study, the MiniQ was relatively simple to apply, and it provided large amounts of information related to brainwave values at the 12 different sites. More specifically, the EEG record of the 10-year-old subject showed lower levels of beta activity in the frontal regions and a higher level of theta activity in the frontal and central regions. However, the slow wave (theta) should be higher in posterior regions and fall in the central area, whereas the fast waves (beta and beta3) should be higher in the anterior regions and lower in the posterior. The subject’s theta/beta ratio was high (Cz: 2.05) and close to values seen in subjects with ADHD according to Monastra et al. [7]. and Jarrett et al. [27]. Although the theta/beta ratio was not high enough to clearly or exactly indicate the presence of ADHD with predominantly inattentive presentation, it is important to consider the full set of data provided by the MiniQ. It is also important to note that the diagnosis of ADHD was reported by the neuropediatrician, who usually uses behavioral criteria. At the same time, we cannot ignore the fact that the use of the theta/beta ratio has also been questioned by other works (e.g., [28]). In any case, the importance of the brainwave analysis lay in helping decide which intervention protocols to follow, along with the frequencies and the sites to use. The chosen neurofeedback protocols were the SMR protocol and the theta/beta protocol. There were 75 intervention sessions, 45 SMR at Cz and 30 theta/beta at Fz. Once the intervention was complete, the changes in theta, beta, beta3 and SMR waves were assessed using the MiniQ.

The intervention produced a variety of results. Firstly, there was a small reduction in theta activity and an increase in SMR, which would indicate better levels of attention. In addition, the theta/beta ratio fell to levels which were closer to those in subjects without ADHD. However, this improvement in the theta/beta ratio was due to increased beta rather than by the reduction of theta. Janssen et al. found similar results in 38 children with ADHD by analyzing the learning curve during 29 neurofeedback training sessions [29]. Their results indicated that while theta activity did not change over the course of the sessions, beta activity showed a linear increase during the study. In our study, the subject was able to significantly improve the levels of beta, but was hardly able to reduce theta activity, which is what would allow even greater improvements in attentional ability. Given this progress, the use of a protocol for inhibition of theta waves at Fz may be effective in strengthening the development of attention levels. Although there were no notable changes at other sites, such as F3 and F4, it is important to note that the intervention was carried out only at Cz and Fz.

On similar lines, during tasks after the intervention (reading and arithmetic), there was no reduction in theta but there was an increase in beta and beta3, again in line with the results from Janssen et al. [29]. For reading and arithmetic, one would expect, at least in subjects without ADHD, that in the frontal regions, values of slow waves would fall and fast waves would rise. However, in this study, there was no increase in beta waves in frontal regions during the tasks. This may indicate that although the neurofeedback intervention protocols in subjects with ADHD produce improvements in baseline activation (increased beta), the same does not happen with activation during the execution of tasks such as reading and arithmetic. In addition, Monastra et al. [7] showed that the activation profile of subjects with ADHD was similar with no task and during a reading task (unlike the control subjects, in whom activation increased during the reading task). Although this fact may be related to the ADHD profile, in our case study, with 75 neurofeedback sessions, we found no differences in the activation of frontal areas during a specific task, such as reading or mathematics.

As Enriquez-Geppert et al. [24] and Duric et al. [25] state, it is still necessary to develop specific procedures (which consider electrode placement and the specific theta/beta, SMR or slow cortical potential protocol) for intervention tailored to the different cases that professionals may find in clinical practice, in order to achieve better results. In this regard, it would be interesting to study theta/beta-ratio learning curves during intervention with neurofeedback, with the aim of achieving better results and making this tool as adaptive as possible in the future.

## 5. Conclusions

These results point toward the hypothesis that the low baseline cortical activation seen in subjects with ADHD would be found to be the basis of the disorder. While neurofeedback training may produce a positive progression, difficulties would persist, particularly during specific tasks in which subjects with ADHD are unable to achieve an ideal profile of brainwave activity for optimum performance. This is a reflection of the fact that the disorder persists throughout life, and hence, despite improvements in the cortical activation profile and the subject learning to strengthen their beta wave activity to concentrate, there will continue to be high levels of theta.

In this context, various studies such as Doppelmayr and Weber [30] and Vernon et al. [31] have reported the benefits of the SMR protocol and others, such as Arns et al. [13], Gevensleben et al. [32] and Leins et al. [33], have done the same with regard to the theta/beta protocol. However, other studies, such as Cortese et al. [34] and Logemann et al. [35], have not found improvements following neurofeedback intervention in children with ADHD. Considering these differences between previous studies, it would be interesting to establish the benefits of one or other of the protocols in interventions in children with ADHD. For example, in adults without ADHD symptoms, Doppelmayr and Weber [30] examined the efficacy of the theta/beta and SMR protocols. They found that the subjects who followed the SMR protocol were able to modulate their brain activity, whereas the theta/beta protocol did not provide benefits in regulation of brain activity.

It is also worth noting that, while previous studies employed similar protocols (SMR, theta/beta), the numbers of sessions and the session durations varied between studies. These variations may be related to the differences in the results and indicate the need to establish intervention protocols not only about what to work with (brain waves) but also how to do it (e.g., number of sessions, session duration, break schedules, etc.). At the same time, the present study underscores the need to tailor protocols to subjects’ profiles, along the same lines as previous studies, for instance Cueli et al. [16], who noted differences in the benefits of interventions based on the type of ADHD presentation. As authors such as Leins et al. [33] have indicated, most neurofeedback intervention programs combine two protocols, and it would be interesting to determine whether the combination is more effective than applying a single protocol.

In the future, it would be advisable to assess subjects’ levels of activation every 10 to 15 sessions of neurofeedback training in order to tailor the protocols to their progress and to study the theta/beta ratio learning curve as mentioned above. One limitation it is important to note is that multidomain assessments before, during, and after treatment (and adequate follow-up) should include blinding and sham inertness Another limitation of the present study is the lack of a behavioral assessment that would allow for an in-depth analysis of the subject’s progress in line with the protocol from Holtmann et al. [36]. At the same time, in spite of the limitations associated with case studies, such as not being able to produce generalizable results, the present work aims to be of some use to clinical and educational professionals so that they may consider intervention protocols for cases similar to the one described here.

Finally, despite the limitations described above, it would also be useful to consider the possibility of incorporating this type of training in more cases of subjects with ADHD, because neurofeedback intervention may offer long-term benefits in terms of improving the attentional abilities of subjects with ADHD, especially if one considers that approximately a third of ADHD patients do not respond to, or sufficiently tolerate, pharmacological treatment [37]. In this regard, it would be interesting to analyze the efficacy of new potential tools that combine neurofeedback and virtual reality and incorporate them into clinical practice [38].

## Figures and Tables

**Figure 1 ijerph-19-00191-f001:**
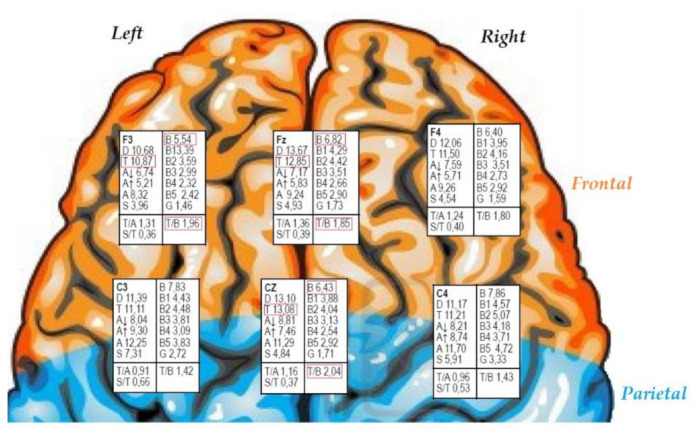
Pre-treatment results from the MiniQ instrument. *Note*. T = theta; B = beta; T/B = theta/beta ratio. In subjects aged between 7 and 11 years old, values over 2.8 for the theta/beta ratio are compatible with a profile of ADHD.

**Figure 2 ijerph-19-00191-f002:**
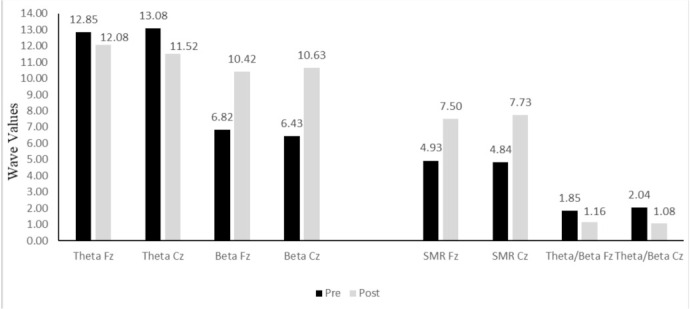
Pre- and post-treatment activity in sites Cz and Fz.

**Figure 3 ijerph-19-00191-f003:**
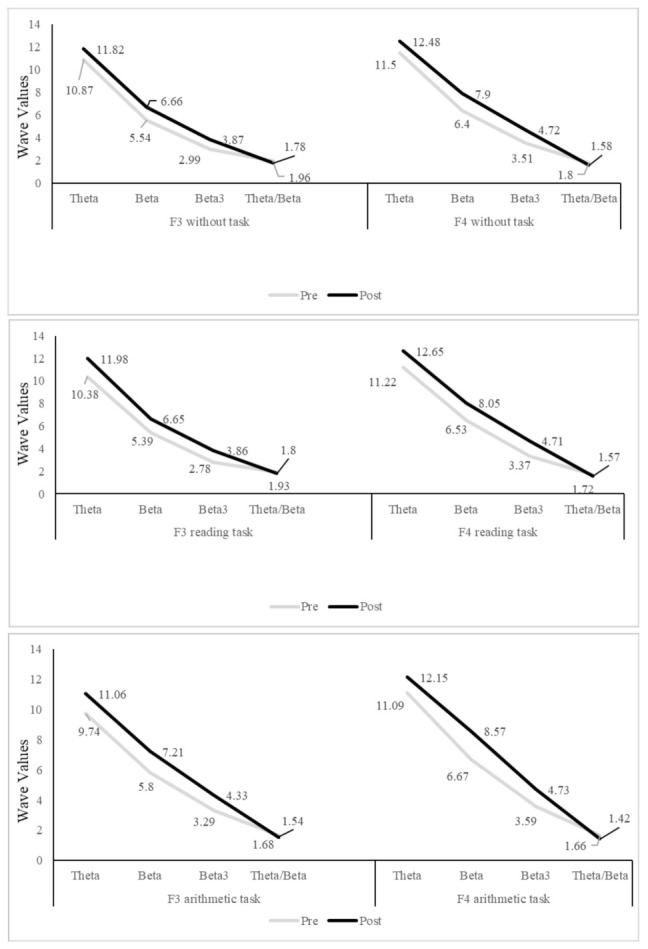
Pre- and post-treatment evolution in F3 and F4 areas with and without tasks.

## Data Availability

The data presented in this study are available on request from the corresponding author.

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
