# Peer review of "A Case Study in Attention-Deficit/Hyperactivity Disorder: An Innovative Neurofeedback-Based Approach"

_ijerph, 2021, doi:10.3390/ijerph19010191_

Round 1

Reviewer 1 Report

This case study revealed the efficacy of neurofeedback approach in the treatment of ADHD. This paper was written clearly, so I don't have any major concerns. Here are some of my minor concerns and suggestions:

  1. Please improve the resolution of Fig1. The numbers and labels are hard to see.
  2. In addition to the pretest and posttest activity results, are there any behavioral assessment to demonstrate the recovery of the subject?
  3. Please add some description to illustrate the difference between the absolute power and relative power of theta/beta ratio.
  4. The pretest theta/beta ratio of this subject is lower than ADHD cutoff. Are there any potential explanations? Otherwise,  the use of this parameter may be questionable, since it is not sensitive. 

Author Response

Dear Reviewer,

Firstly, we would like to thank you for the time taken. For us, all the reviews and recommendations have been enormously helpful. The current version is not only more readable and clearly, but also it has increased its scientific quality.

Best,

Authors.

REVIEWER 1

Reviewer 1: This case study revealed the efficacy of neurofeedback approach in the treatment of ADHD. This paper was written clearly, so I don't have any major concerns. Here are some of my minor concerns and suggestions

  • Response: Thank you for your comment.

Reviewer 1: Please improve the resolution of Fig1. The numbers and labels are hard to see.

  • Response: The resolution of the figure 1 has been improved. We reduced the image in order to increase quality.

Reviewer 1: In addition to the pretest and posttest activity results, are there any behavioral assessment to demonstrate the recovery of the subject?

  • Response: We have included as limitation of the present study the lack of a behavioral assessment that would allow for an in-depth analysis of the subject's evolution in line with the protocol of Holtmann et al. [34].

Reviewer 1: Please add some description to illustrate the difference between the absolute power and relative power of theta/beta ratio.

  • Response: Different authors establish different normative values depending on the amplitudes of the beta or theta waves, the important thing is that the amplitude of the wave is the same. Biograph uses ranges of theta 4-8 hz and beta 13-21 hz, which are the same as those used by Lubar and Monastra. We have included in the manuscript (subsection 2.2.1.) that traditional ratios for ADHD indicators use absolute power values measured in peak volts (microvolts squared by the hertz value). Biograph for theta/beta ratio calculation uses relative power values (microvolts squared by the hertz value).

Reviewer 1: The pretest theta/beta ratio of this subject is lower than ADHD cutoff. Are there any potential explanations? Otherwise, the use of this parameter may be questionable, since it is not sensitive.

  • Response: We have tried to justify in the literature the use of theta/beta ratio. Also, we have added in the discussion a reflection in relation with the cutoff points used and the need of update the values considering age and sex. At the same time, we have highlighted that the relevance of the brainwave analysis rested in the possibility of deciding which intervention protocol to follow, along with the frequencies and the sites to use

Reviewer 2 Report

Among nonpharmacological treatments for ADHD identified neurofeedback is one of the more promising treatments, with evidence supporting sustained benefit after 6 to 12 months. The treatment package called neurofeedback is likely multifactorial, including primary reinforcement of targeted neurophysiological activity, reinforcement by psychological factors implicit in treatment protocols, placebo response, and synergism with other treatments (psychotherapy, coaching, sleep hygiene, etc).

Since 1976 studies were published and results of randomized clinical trials (also multicenter) were produced.

Thus, what the present case-study adds? What is the message for the practice?

Authors should better contextualize their work and discuss in more detail how "These instruments may be beneficial in the evaluation and treatment of ADHD".

Multidomain assessments before, during, and after treatment (and adequate follow-up) should be include tests of blinding and sham inertness. 

In the present form the text is too long and in som eplaces generic.

References should be updated and adequately discussed.

Author Response

Dear Reviewer,

Firstly, we would like to thank you for the time taken. For us, all the reviews and recommendations have been enormously helpful. 

Best,

Authors.

Reviewer 2: Among nonpharmacological treatments for ADHD identified neurofeedback is one of the more promising treatments, with evidence supporting sustained benefit after 6 to 12 months. The treatment package called neurofeedback is likely multifactorial, including primary reinforcement of targeted neurophysiological activity, reinforcement by psychological factors implicit in treatment protocols, placebo response, and synergism with other treatments (psychotherapy, coaching, sleep hygiene, etc). Since 1976 studies were published and results of randomized clinical trials (also multicenter) were produced. Thus, what the present case-study adds? What is the message for the practice?

  • Response: Page 3. We have included that following to Duric et al. [25], there is no standard recommended regarding the number, time and frequency of sessions, and standard placement of NF screening when this type of protocols are administered. In this context, the present study aims to provide a structure in which the neurofeedback intervention is adjusted to the data provided by the previous assessment.

Reviewer 2: Authors should better contextualize their work and discuss in more detail how "These instruments may be beneficial in the evaluation and treatment of ADHD".

  • Response: Thank you for your comment. We have included the implication of the present study. Also, we have reduced some generic information in order to specify the aspects in which the work tries to provide an innovation. The instruments used in the present study are not new. However, in relation with the neurofeedback, little is known about, how we have to implement it to obtain the better results.

Reviewer 2: Multidomain assessments before, during, and after treatment (and adequate follow-up) should be include tests of blinding and sham inertness.

  • Response: We have included as limitation of the study, the lack of this kind of measures.

Reviewer 2: In the present form the text is too long and in some places generic.

  • Response: We have reduced generic information.

Reviewer 2: References should be updated and adequately discussed.

  • Response: We have included new references.

Reviewer 3 Report

GENERAL COMMENT
The authors reported a single experimental case with an ADHD child to asses the psychometric properties of. The paper is quite clear and the main finding is that the neurofeedback treatment can improve the theta/beta ratio though the increase go Beta and Beta3 but no reduction in theta activity was registered. This falls to reached a greater improvements in attention ability.
I don’t see an innovative approach in this study, because aimed on a protocol well know and well known and explored in recent decades. 
The recommendation is to accept the paper after a revision taking into account the following specific comments. 
SPECIFIC COMMENTS
1. INTRODUCTION
The way as the reference cited are not easy to read. 
For example, in the subsection 1.2 paper 11 is cited 5 times. Please, rephrase e.i. line 137-138.
 The subsection 1.1 is rich of technical details in line mostly with M&M section. I propose to reduce the subsection in the introduction and add a new on the the methodology section.
2. METHODOLOGY :
Line 211 : Typical SMR protocol used the C3-C4 electrode, where typical theta/beta protocol used Fz-Cz . The author’s choice is focused on the electrode Cz in SMR protocol and only Fz in theta/beta protocol. This choice need to be justified.
Figure 1 is unreadable. Please, increase the resolution to be useful introduce all this data. 
Figure 2 : title in x and y axes need to be added. I suggest to replace Pretest and Posttest with Pre and Post treatment in the figures as well as in the text. 
Figure 3 : title in x axe need to be translate, when title in y axe need to be added. Keep the same colours used for Figure 2, this can be annoying during the reading. Caption is the same as figure 2, please, correct the mistake ! 
3. Complementary cited literature to be added in the introduction and discussion:
[1] Holtmann et al., 2014. http://www.biomedcentral.com/1471-2431/14/202  
[2] Blume  et al., 2017. https://pubmed.ncbi.nlm.nih.gov/28118856/
[3] Enriquez-Geppert  et al., 2019. https://doi.org/10.1007/s11920-019-1021-4 

Author Response

Dear Reviewer,

Firstly, we would like to thank you for the time taken. For us, all the reviews and recommendations have been enormously helpful.

Best,

Authors.

GENERAL COMMENT

Reviewer 3: The authors reported a single experimental case with an ADHD child to assess the psychometric properties of. The paper is quite clear and the main finding is that the neurofeedback treatment can improve the theta/beta ratio though the increase go Beta and Beta3 but no reduction in theta activity was registered. This falls to reached a greater improvements in attention ability. I don’t see an innovative approach in this study, because aimed on a protocol well know and well known and explored in recent decades. The recommendation is to accept the paper after a revision taking into account the following specific comments.

- Response: Thank you for your comment, we have included in the introduction (page 3) the implication of the present study in which it tries to provide an innovation. The instruments used in the present study are not new. However, in relation with the neurofeedback, little is known about, how we have to implement it to obtain the better results. The protocols do not stablish the specific method of implementation.

SPECIFIC COMMENTS

1. INTRODUCTION

Reviewer 3: The way as the reference cited are not easy to read. For example, in the subsection 1.2 paper 11 is cited 5 times. Please, rephrase e.i. line 137-138.

- Response: We have reformulated second and third paragraphs in subsection 1.2.

Reviewer 3: The subsection 1.1 is rich of technical details in line mostly with M&M section. I propose to reduce the subsection in the introduction and add a new on the methodology section.

- Response: We have included information of the subsection 1.1 in the methodology (subsection 2.2.1).

2. METHODOLOGY:

Reviewer 3: Line 211: Typical SMR protocol used the C3-C4 electrode, where typical theta/beta protocol used Fz-Cz. The author’s choice is focused on the electrode Cz in SMR protocol and only Fz in theta/beta protocol. This choice need to be justified.

- Response: Lines 211 and 255. We have justified the sites used for the intervention based on the results of the previous evaluation. Also, we have included that typical SMR protocol sued

Reviewer 3: Figure 1 is unreadable. Please, increase the resolution to be useful introduce all this data.

- Response: The resolution of the figure 1 has been improved. We reduced the image in order to increase quality.

Reviewer 3: Figure 2: title in x and y axes need to be added. I suggest to replace Pretest and Posttest with Pre and Post treatment in the figures as well as in the text.

- Response: We have added the title in x and y axed. We have used the terms Pre and Post treatment in the manuscript.

Reviewer 3: Figure 3: title in x axe need to be translate, when title in y axe need to be added. Keep the same colours used for Figure 2, this can be annoying during the reading. Caption is the same as figure 2, please, correct the mistake!

- Response: Figure 3 has been modified including the title in x axe in English and adding the title in y axe. Also, the same colours than in figure 2 are used. The title of the figure has been corrected.

Reviewer 3: Complementary cited literature to be added in the introduction and discussion:
[1] Holtmann et al., 2014. http://www.biomedcentral.com/1471-2431/14/202
[2] Blume et al., 2017. https://pubmed.ncbi.nlm.nih.gov/28118856/
[3] Enriquez-Geppert et al., 2019. https://doi.org/10.1007/s11920-019-1021-4

- Response: We have included the suggested references.

Round 2

Reviewer 2 Report

The authors tried to improve the presentation and in part they succeeded. The limits remain, as too them know.

Author Response

Dear Reviewer,

Thank you again for the opportunity to revise our work. We are aware of the limitations of the work and have tried to reflect these limitations in the corresponding section of the manuscript. In addition, after this new review, we have incorporated more specific aspects in the discussion that we consider have improved this section. 

Sincerely,

The authors,